# SARS-CoV-2 Infection-Blocking Immunity Post Natural Infection: The Role of Vitamin D

**DOI:** 10.3390/vaccines11020475

**Published:** 2023-02-17

**Authors:** Rami Abu Fanne, Mahmud Moed, Aviv Kedem, Ghalib Lidawi, Emad Maraga, Fady Mohsen, Ariel Roguin, Simcha-Ron Meisel

**Affiliations:** 1Leumit Health Services, Tel Aviv 6473817, Israel; 2Heart Institute, Hillel Yaffe Medical Center, Hadera 3810101, Israel; 3Urology Department, Hillel Yaffe Medical Center, Hadera 3810101, Israel; 4Clinical Biochemistry Department, Hadassah Medical Center, Jerusalem 9103102, Israel

**Keywords:** vitamin D, humoral response, reinfection, recovery

## Abstract

Objective and Aim: The extent of the protection against SARS-CoV-2 conferred by natural infection is unclear. Vitamin D may have a role in the interplay between SARS-CoV-2 infection and the evolving acquired immunity against it. We tested the correlation between baseline 25(OH) D content and both the reinfection rate and the anti-spike protein antibody titer following COVID-19 infection. Methods A retrospective observational survey that included a large convalescent COVID-19 population of subjects insured by the Leumit HMO was recorded between 1 February 2020 and 30 January 2022. Inclusion criteria required at least one available 25(OH)D level prior to enlistment. The association between 25(OH)D levels, the rate of breakthrough infection, and the anti-spike protein antibody titer was evaluated. Results A total of 10,132 COVID-19 convalescent subjects were included, of whom 322 (3.3%) sustained reinfection within a one-year follow-up. In the first 8 months after recovery, the reinfected patients were characterized by a higher incidence of low 25(OH)D levels (<30 ng/mL, 92% vs. 84.8%, *p* < 0.05), while during the following three months, the incidence of low 25(OH)D levels was non-significantly higher among PCR-negative convalescent subjects compared to those reinfected (86% vs. 81.7, *p* = 0.15). By multivariate analysis, age > 44 years (OR-0.39, 95% CI: 0.173–0.87, *p* = 0.02) and anti-spike protein antibody titer > 50 AU/mL (0.49, 95% CI: 0.25–0.96, *p* = 0.04) were inversely related to reinfection. No consistent correlation with vitamin D levels was observed among the 3351 available anti-spike protein antibody titers of convalescent subjects. However, the median anti-spike protein antibody titers tended to increase over time in the vitamin D-deficient group. Conclusion Higher pre-infection 25(OH)D level correlated with protective COVID-19 immunity during the first 8 months following COVID-19 infection, which could not be explained by anti-spike protein antibody titers. This effect dissipated beyond this period, demonstrating a biphasic 25(OH)D association that warrants future studies.

## 1. Introduction

Since the advent of the COVID-19 pandemic at the beginning of 2020, there has been a worldwide effort to cope with and protect the global population against its negative, multifaceted impact. Despite unprecedented swift development and application of novel vaccines, and the rise in naturally occurring immunity, there are still parts of the world where the epidemic is rampant, and others where individuals are commonly reinfected with COVID-19. This apparently occurs as a result of failure in obtaining durable immunity against infection. Immunity, whether natural following infection or due to vaccination, seems to wane with time, thus limiting long-term protection [1,2,3,4,5]. In this complex landscape, there is a growing debate regarding the efficacy and durability of immune memory in convalescent patients compared to that in fully vaccinated individuals. While most studies report non-inferiority of natural immunity relative to vaccine-induced immunity, recent reports have claimed the superiority of the former [6,7,8,9,10,11]. However, the long-lasting protective effect of both types of immune induction is limited by the mutability of COVID-19, allowing evasion of the human immune system [12].

A potential complementary approach to achieve a robust immune response, irrespective of the strategy adopted or the variant encountered, is to take advantage of potential immune adjuvants, primarily vitamin D [13]. Immunomodulation by vitamin D might impact the pre-infection and the infection stages, as well as the post-infection and vaccination stages. In the pre-infectious and infectious stages, 25(OH)D deficiency correlated with increased infection rate and COVID-19-related complications including death [14,15]. Nevertheless, studies assessing the optimization of COVID-19 outcomes through vitamin D supplementation yielded equivocal results; vitamin D substitution reduced cough duration [16], shortened hospital stay, and decreased mortality among patients with COVID-19 infection [17]. In contrast, a single high dose of vitamin D3 administered to hospitalized patients with moderate–severe COVID-19 disease showed no effect [18]. Reports from the acquired immunity stage have recently been provided [19], showing a null effect of vitamin D supplementation on the protective efficacy or immunogenicity of SARS-CoV-2 vaccination.

We describe here the results of a large population-based data study evaluating the association between baseline plasma 25(OH)D content and both the anti-spike antibody levels and the reinfection rate among SARS-CoV-2 recovered subjects.

## 2. Methods and Patients

We conducted a population-based study among adult members of Leumit Health Services (LHS), a large Israeli nationwide health maintenance organization (HMO), which provides health services to nearly 730,000 members. LHS has a comprehensive computerized database, continuously updated regarding the demographics, medical diagnoses and clinic visits, hospitalizations, and laboratory tests of insured members.

The socio-economic status (SES) was defined according to the home address. The Israeli Central Bureau of Statistics categorizes all cities and settlements into 20 SES levels. Classification at levels 1–9 is considered low–medium SES, while levels 10–20 represent the medium–high SES. Ethnicity was also defined according to the home address of the HMO members, and categorized into three groups: general population, ultra-orthodox Jews, and Arabs.

All LHS members have identical health insurance coverage and access to healthcare services. Relevant diagnoses are entered or updated according to the International Classification of Diseases 10th revision (ICD-10). The validity of chronic diagnoses in the registry has been previously established (Hamood et al., 2016; Rennert and Peterburg, 2001). The study population included all LHS members aged 18 or older who fulfilled the following criteria:

Recovery from documented COVID-19 infection between 1 February 2020, to 30 January 2022, in the absence of prior vaccination;

At least one plasma 25(OH)D level prior to infection and recruitment;

RT-PCR test for SARS-CoV-2 performed ≥ 3 months after recovery, and before booster injection if any.

We extracted available SARS-CoV-2 serology and associated demographic and clinical data for all study subjects. SARS-CoV-2 RT-PCR testing after recovery followed the Israeli Ministry of Health instructions of performing COVID-19 testing indicated upon exposure to confirmed COVID-19 patients or in the presence of symptoms suggestive of COVID-19 infection. The Allplex 2019-nCoV assay (Seegene, Seoul, Republic of Korea) was used until 10 March 2020, followed by employment of the COBAS SARS-CoV-2 6800/8800 assay (Roche Pharmaceuticals, Basel, Switzerland). Regarding serological testing, referrals to SARS-CoV-2 IgG testing were left to the discretion of the treating physician. Test results were not intended to determine the need for vaccination. The Abbot Alinity™ i system (Illinois, IL, USA) was employed for antibody assay. The Abbott Alinity™ system showed reliable results by internal testing with 99.6% specificity and 100% sensitivity for COVID-19 patients tested 14 days after the initial symptoms [20]. The Abbott assay was validated externally with excellent sensitivity and specificity [21]. Qualitative results and index values reported by the system were used in the analyses.

Baseline medical conditions known to be associated with the severity of COVID-19 infection or the antibody level in the adult population, including obesity, diabetes mellitus, hypertension, asthma, chronic obstructive pulmonary disease, ischemic heart disease, the presence of malignancy, and chronic kidney disease, were recorded. Obesity was defined as BMI > 30 kg/m^2^. According to LHS guidelines, vitamin D tests were collected after overnight fasting and transported on ice to the central laboratory for processing within 4 h of collection using the DiaSorin Chemiluminescence assay [22,23,24,25]. For categorization of vitamin D levels, the common convention of most scientific societies was adopted, with values lower than 20 ng/mL representing vitamin D deficiency, concentrations of 21–29 ng/mL considered insufficient, and values > 30 ng/mL reflecting adequate levels. The study protocol was approved by the LHS Institutional Review Board (13-21-LEU).

### Statistical Analysis

Descriptive statistics in terms of mean, standard deviation, median, and percentiles were presented for all parameters in the study. Differences between groups (positive PCR vs. negative PCR, antibody titer < 50 vs. antibody titer ≥ 50) were presented by *t*-test or Fisher exact tests for continuous and categorical parameters, respectively. Differences within groups (positive PCR or negative PCR) according to vitamin D levels and time interval from the second vaccination were calculated with Pearson Chi-square. As the distributions of the antibody levels were not normally distributed (by Kolmogorov–Smirnov test), we used the Log-transformed function. Multi-level assessments of vitamin D, PCR status, antibody levels, and the time interval elapsed from recovery were calculated using Kruskal–Wallis tests with multiple comparisons. Parameters were selected as candidates for the multivariate analysis based on their significance from the univariate analysis. The multivariate logistic regression model was assessed to determine the effect of the independent parameters associated with positive PCR. A *p*-value < 0.05 was considered significant. IBM^®^SPSS version 28 was used for all statistical analyses.

## 3. Results

During the study period, 28,605 convalescent subjects with available baseline plasma 25(OH)D levels, and without previous vaccination were identified. Of this group, 10,132 subjects who fulfilled the inclusion criteria comprised the study cohort. The flow diagram used for cohort selection is displayed in Figure 1, while their characteristics at inclusion are shown in Table 1. Notably, the median interval that elapsed from the time of 25(OH)D measurement to the time of the first documented infection was 4 months (IQR: 1–7). As mentioned before, all of the trial’s participants had an RT-PCR test for SARS-CoV-2 performed ≥ 3 months after recovery. Overall, the reinfection rate was 3.3% at a median follow-up of 8.6 months (6.2–11.3). In other words, 96.7% of the study population were RT-PCR negative at a median of 6.7 months (3.8–9.3).

Primary univariate analysis showed that younger age (*p* < 0.0001), Arab ethnicity (*p* < 0.0001), asthma (*p* = 0.018), low socioeconomic status (*p* < 0.001), BMI < 25 (*p* = 0.004), and anti-spike protein antibody titers < 50 AU/mL (*p* = 0.051) were positively associated with COVID-19 reinfection. Interestingly, a history of CVA (*p* = 0.045) and hyperlipidemia (*p* < 0.001) were negatively linked with reinfection. Figure 2 illustrates the association between vitamin D levels and the PCR status (positive or negative) at different time intervals following recovery. During the first 8 months after recovery, the PCR-positive group, i.e., the reinfected subjects, were characterized by a higher incidence of low 25(OH)D (< 30 ng/mL) levels (95% vs. 84% at 3–4 months, 90% vs. 85% at 5–6 months, 91% vs. 85% at 7–8 months, and overall, 92% vs. 84.8% between 3–8 months after recovery, *p* < 0.05); however, during the 9–12 month period, the trend was reversed with a tendency toward a higher incidence of low 25(OH)D level among non-reinfected subjects (81.7% vs. 86%, *p =* 0.15). Of note, the number of subjects tested in each period clearly shows that the sample size of SARS-CoV-2 RT-PCR testing during the 9–12-month period was accounting for 29% (2990 participants) of the total tests performed (Figure 2), sufficiently powered to reflect the population of interest.

The infection rate at each specific period as related to the vitamin D level category is shown in Table 2. Up to months 7–8, lower vitamin D levels (<30 ng/mL) were associated with a higher infection rate. The overall infection rate during the first 8 months post-recovery was 3.34%, 2.54%, 1.53%, and 3.03% at vitamin D levels of <20, 20–30, > 30, and < 30 ng/mL, respectively. We then performed a statistical significance testing between the infection rate at the different vitamin D levels, showing significant differences between sufficient vitamin D levels (>30 ng/mL) and both low (<30 ng/mL) and deficient (<20 ng/mL) vitamin D levels: 1.53% vs. 2.54%, *p* = 0.07; 1.53% vs. 3.34%, *p* = 0.0012; 2.54% vs. 3.34%, *p* = 0.07; 1.53% vs. 3.03%, *p* = 0.002. Interestingly, during the 9–12-month interval after recovery, the correlation between the infection rate and vitamin D levels demonstrated an opposite trend, i.e., more infections documented with sufficient (>30 ng/mL) vitamin D levels: the infection rate correlated numerically; however, it did not correlate significantly with the brackets of vitamin D levels (*p* = NS).

A multivariate regression model applied after controlling for demographic variables and comorbidities showed a significant negative association between both age > 44 years (0.387, 95% CI: 0.17–0.87, *p* = 0.021), anti-spike protein antibody titer > 50 AU/mL (0.49, 95% CI: 0.25–0.97, *p* = 0.039) and the likelihood of reinfection.

Of the 3351 recovered patients who had available anti-spike antibody tests, 3248 patients (96.9%) had a negative RT-PCR test while 103 (3.1%) were reinfected. Tracking the anti-spike antibody levels as a function of the time elapsed from recovery showed that, overall, the values remain constant or level off over time (Figure 3). A linear regression model fitted to quantify the association between the elapsed time from recovery and the anti-spike antibody levels yielded the equation 676 + 66.7× time (in months). The equation illustrates the overall stability of titers, with a non-significantly measured increase in total anti-spike antibody levels over time. Next, we evaluated the association between the median anti-spike antibody titers and the time elapsed from recovery stratified by baseline plasma 25(OH)D levels (Figure 4). Deficient vitamin D levels (<20 ng/mL) during the 5–12-month interval after recovery were associated with non-significantly higher anti-spike antibody titers, while baseline vitamin D levels > 20 ng/mL were associated with steady antibody levels. No significant difference in median anti-spike antibody titers was observed between the different 25(OH)D levels at all time intervals.

In an attempt to assess factors associated with anti-spike antibody level, study subjects were divided into two groups according to anti-spike antibody titer (lower or higher than 50 AU/mL). The median time elapsed from recovery to anti-spike antibody sampling was 5.4 months (IQR: 3.6–7.3). The characteristics of each subset are presented in Table 3. No single factor was a significant predictor of anti-spike antibody titer, including plasma 25(OH)D level itself.

Altogether, we found a significantly higher proportion of patients with anti-spike antibody < 50 AU/mL among positive RT-PCR subjects (44%) compared to a lower incidence of positive RT-PCR observed among subjects with anti-spike antibody titers > 50 AU/mL (30%, *p* < 0.05). Accordingly, the incidence of reinfection was 4.5% among subjects with anti-spike antibody < 50 AU/mL level compared to an incidence of 2.5% in the presence of anti-spike antibody titer > 50 AU/mL (*p* = 0.02).

## 4. Discussion

The mainstay of prevention of COVID-19 infection is the immunological memory acquired after vaccination or previous COVID-19 infection. Unfortunately, more than 30 months into the pandemic prove that immunity is short-lived, mainly the result of the mutable nature of COVID-19 and the rapid waning of acquired immunity [1,2,3,4,5]. In this context, the ancillary role of 25(OH)D as an immune modulator has repeatedly been suggested. We used a data set involving 10,132 COVID-19 convalescent individuals from an integrated healthcare organization to evaluate a potential association between pre-infection 25(OH)D serum levels and both the anti-spike antibody titer, representing humoral immunity, and the reinfection rate.

A protective association between high vitamin D levels and the primary prevention of COVID-19 infection has previously been reported [26]. A valuable finding of the current study is the relevance of 25(OH)D in the prevention of reinfection in the first 8 months post recovery. The recovered cohort virtually represents a homogeneous group of past RT-PCR-positive patients, a population already linked with lower vitamin D levels [26]. Interestingly, despite being a vulnerable cohort with established lower 25(OH)D contents, the reinfected subjects in the current study still displayed lower 25(OH)D levels relative to the non-reinfected group, at least during the first 8 months after recovery.

In accordance with a previous study from LHS [27] before the launch of COVID-19 vaccination, we found older age to be inversely correlated with reinfection in convalescent subjects. Older age was associated with higher adherence to COVID-19 preventive measures, including social distancing [28]. This behavior may explain the low infection rate in older subjects. Similarly, anti-spike protein antibody titer > 50 AU/mL was associated with a lower risk of infection. Most of the recovered group sustained minor symptoms during reinfection, explaining the restrained humoral response, with lower initial anti-spike antibody titers (months 3–4) associated with lower vitamin D levels (*p* = NS). Over time, the median antibody titers increased in the deficient vitamin D group, while a restrained increase was observed in subjects with higher 25(OH)D levels. In this context, several studies have reported an inverse relationship between serum 25(OH)D and virus antibody titers [29,30,31]. Mechanistically, vitamin D has been shown to hamper the production of immunoglobulins [32,33,34,35], attenuating the humoral immune reaction induced by viruses. The continued positive antibody evolution in the long term may reflect the frequent occurrence (30–60%) of long-COVID-19 cases [36], presumably due to small amounts of SARS-CoV-2 antigen or lack of complete viral clearance [37].

Despite minimal heterogeneity in the magnitude of antibody titers during the first 8 months post-recovery in the different vitamin D categories (Figure 4), we found inferior protection against reinfection with deficient vitamin D levels during this period. This observation indicates that a potential protective effect of vitamin D at this stage could not be mediated by immunoglobulins; nevertheless, the association between low vitamin D and reinfection seems to be reversed or absent during the 9–12-month period. As mentioned above, late convalescence was dominated by higher anti-spike protein antibody titer in the low vitamin D (0–20 ng/mL) subjects. The heightened humoral response at this stage might explain, at least partially, the mitigated reinfection risk.

Although the vast majority of tests with anti-spike protein antibody titer < 50 AU/mL (73.9%) occurred in patients with low serum 25(OH)D levels, we did not find a significant association between baseline 25(OH)D levels and the proportion of anti-spike protein antibody titers < 50 AU/mL, and at each time point we observed nearly 30% of anti-spike protein antibody titers < 50 AU/mL, the so-called “non-responders” (Figure 5). Of note, the sufficient 25(OH)D group showed the highest percentage of tests with anti-spike protein antibody titers < 50 AU/mL in the long term, once again demonstrating a potential hampering effect of vitamin D on antibody production.

The overall reinfection rate at 12 months after infection was only 3.2% in the current convalescent group, with a documented surge after 10 months from recovery. Interestingly, the breakthrough infection rate during the same period among post-vaccinated (previously non-infected) individuals was 6.9% [34], despite higher baseline vitamin D levels and substantially higher levels of measured anti-spike antibodies, and the surge in breakthrough infection was documented already at 5–6 months after vaccination. The above observations prove the complexity of the immunity puzzle. Vitamin D and anti-spike antibodies are simply just two measurable pieces of the puzzle. Nevertheless, we observed a substantial anti-spike antibody decay in the vaccinated cohort over time, as opposed to sustained levels in the naturally infected cohort. Acknowledging that anti-spike antibodies are not a direct measure of the immune neutralizing capacity, we surmise that the protective association of the maintained levels of anti-spike antibody titer in the convalescent cohort is a correlate of sustained, multi-faceted immunological response. The missing pieces of the puzzle within the complex framework of natural immunity are cellular immunity and other antibody types targeting multiple epitopes. In this regard, T cells seem to possess a pivotal role in SARS-CoV-2 protective immunity [38,39]. The advantage of targeting different immunogenic epitopes other than the spike protein during natural infection was reported before [40,41]; hence, targeting other immutable components of the viral structure, or multiple parts of the virus are emerging exciting avenues for vaccine optimization.

## 5. Study Limitations

Some study limitations require mentioning. First, the history of vitamin D supplementation was not accessible in the present study. In this regard, the Health Ministry published updated nutritional guidelines for the entire population in April 2020, recommending the consumption of a daily vitamin D supplement of 800–1000 IU. As a vitamin D supplement is a common over-the-counter purchase, the retrospective study design could not possibly assess adherence to this recommendation. Second, the current study is of a cross-sectional design providing point values from different individuals collected at different time points after COVID-19 infection. Optimal prognostication of immunity requires a longitudinal study design using multiple sequential samples from individuals in the convalescent phase to better understand the kinetics, magnitude, and durability of antibodies. Moreover, the magnitude of serological immune responses to SARS-CoV-2 infection is highly variable, with differences attributed to differences in antigen exposure, variable viral load trajectories, disease severity, patient age, and comorbidities. Of note, most of these factors might be mitigated in the context of uniform vaccination. Lastly, it is important to keep in mind that many anti-spike antibodies are not neutralizing, and their total titer is a correlate, rather than a direct measure, of neutralizing antibodies [42]. Previous studies have shown progressive decay of neutralizing antibody levels over time, even with maintained levels of serologically measured anti-spike antibodies [43]. Nevertheless, the substantial association in the current study between prior vitamin D deficiency and positive RT-PCR for COVID-19 suggests that most recovered individuals who tested positive for SARS-CoV-2 infection were characterized by low 25(OH)D values when contracting COVID-19. However, this limitation is typical in very large patient studies, where minute individual details, occasionally important, are not available. The rationale behind these large-scale studies is that individual variations tend to mutually cancel each other due to the large number of patients.

## 6. Conclusions

The immune response to COVID-19 is complex, and reliable correlates of protection are ill-established. The current study emphasizes the potential impact of pre-infection levels of vitamin D on reinfection during convalescence and the limitation of the immunoglobulins titer to serve as a stand-alone criterion to predict protective immunity against COVID-19. Finally, the findings underscore the importance of additional unmeasured immune components that operate in this complex process of protection against infection.

## Figures and Tables

**Figure 1 vaccines-11-00475-f001:**
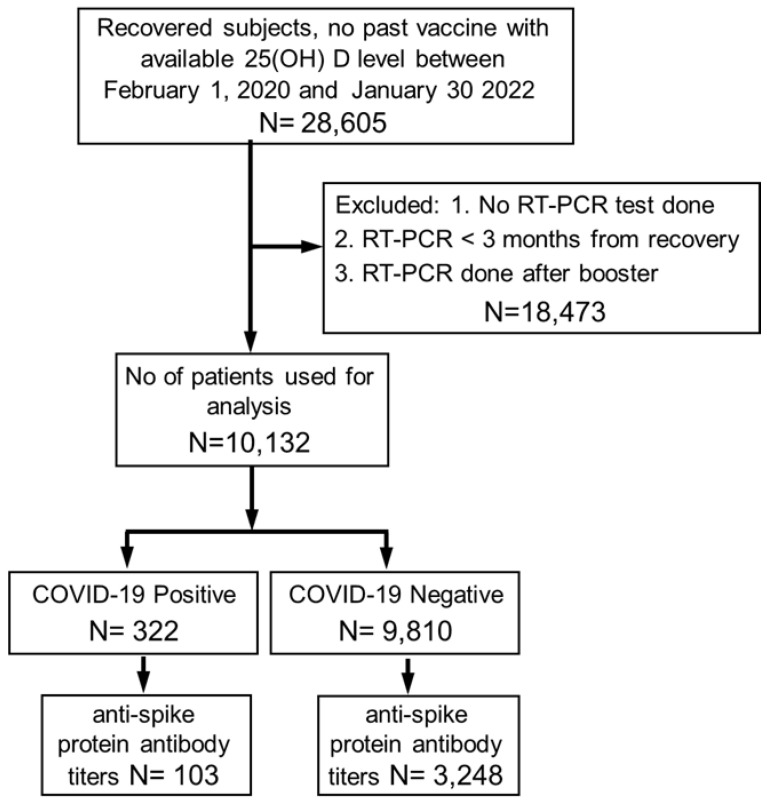
Scheme of the study flowchart.

**Figure 2 vaccines-11-00475-f002:**
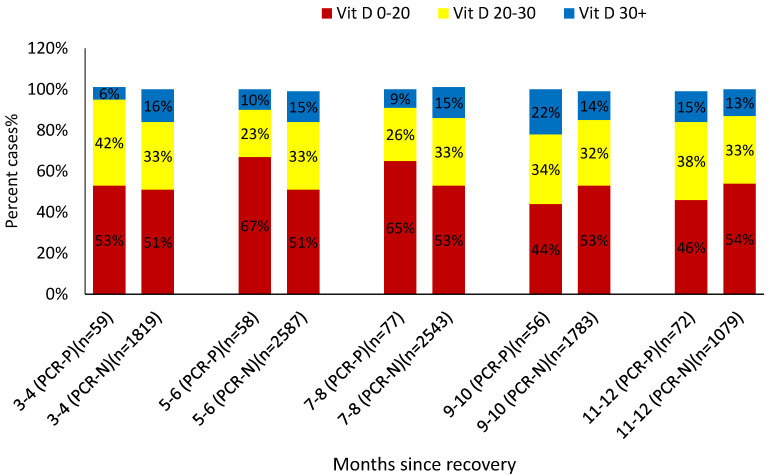
Percent cases with negative/positive RT-PCR classified by baseline 25(OH) D levels at each time point since recovery. PCR-P—PCR Positive; PCR-N—PCR Negative. n-Number of individuals at each period, in the positive and negative PCR results.

**Figure 3 vaccines-11-00475-f003:**
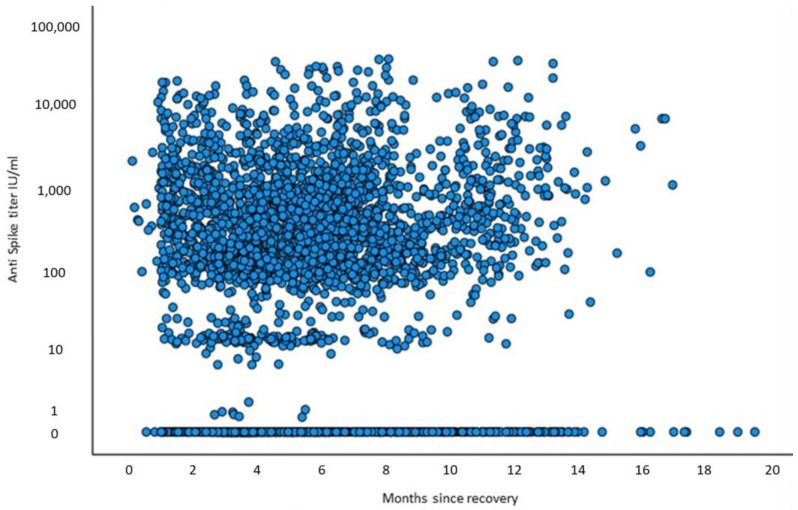
Scatter plot (BIVAR), antibody titers plotted against elapsed time since recovery.

**Figure 4 vaccines-11-00475-f004:**
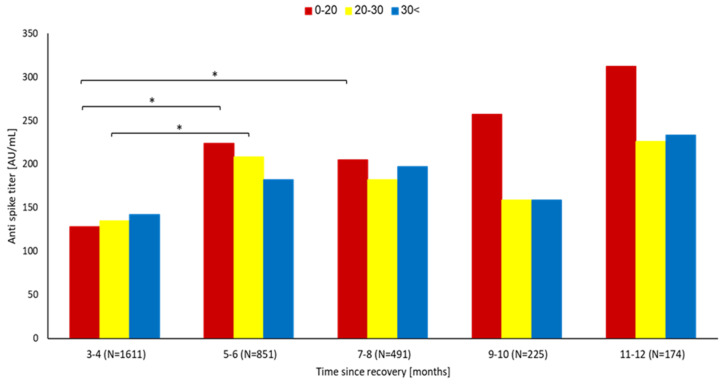
Anti-spike antibody titers of the convalescent group stratified by baseline pre-infection 25(OH) D levels at each 2-month interval since recovery. * *p* < 0.05.

**Figure 5 vaccines-11-00475-f005:**
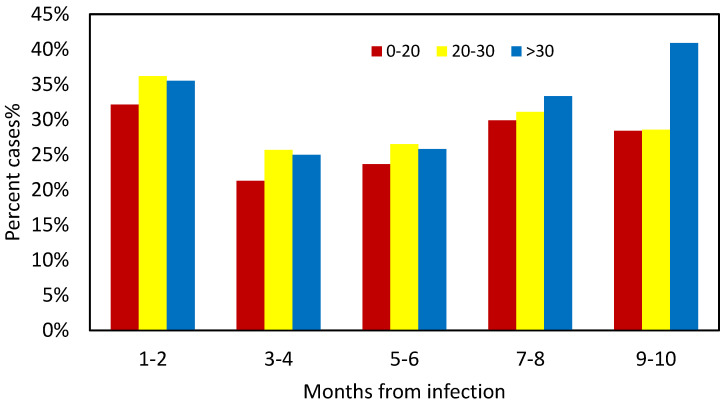
The proportion of anti-spike protein antibody titers < 50 AU/mL among available anti-spike protein antibody tests according to 25(OH) D level at each period post-recovery.

**Table 1 vaccines-11-00475-t001:** Baseline characteristics of the overall cohort by PCR status. CVA—cerebrovascular accident; CHF—congestive heart failure; PVD—peripheral vascular disease; IHD—ischemic heart disease; HTN—hypertension; DM—diabetes mellitus; COPD—chronic obstructive pulmonary disease; SES—socioeconomic state; BMI—body mass index.

	PCR Negative (*n* = 9810)	PCR Positive (*n* = 322)	*p*-Value
Age	40.9 ± 19.1	33.06 ± 16.56	*p* < 0.0001
Gender Male Female	(35.5%) (64.5%)	(34.5%) (65.5%)	*p* = 0.72
Ethnicity Arab Orthodox Jewish Other	(34%) (22%) (44%)	(56.5%) (11.5%) (32%)	*p* < 0.0001
25(OH) D levels 10–20 20–30 >30	(52%) (33%) (15%)	(55%) (33%) (12%)	*p* = 0.36
Anxiety	(38%)	(40%)	*p* = 0.65
Schizophrenia	(3%)	(6%)	*p* = 0.057
Depression	(24%)	(27%)	*p* = 0.53
Dementia	(3.7%)	(1.6%)	*p* = 0.33
Nephrotic syndrome	(0.4%)	(0.8%)	*p* = 0.43
Chronic Renal failure	(5.1%)	(4.7%)	*p* = 1.00
CVA	(4.4%)	(0.8%)	*p* = 0.045
CHF	(3.6%)	(2.4%)	*p* = 0.63
PVD	(3.3%)	(0.8%)	*p* = 0.13
IHD	(7.6%)	(5.5%)	*p* = 0.49
Hyperlipidemia	(49%)	(31.5%)	*p* < 0.001
HTN	(1.5%)	(0.8%)	*p* = 1.00
DM	(11%)	(6.3%)	*p* = 0.11
COPD	(7.3%)	(5.5%)	*p* = 0.60
Asthma	(19.7%)	(28.3%)	*p* = 0.018
SES 1–10 10–20	(67%) (33%)	(76%) (24%)	*p* < 0.001
Smoking status Active smoker Non smoker Former smoker	(11%) (88%) (1%)	(13%) (86%) (1%)	*p* = 0.43
Anti-spike antibody titer <50 ≥50	(30.5%) (69.5%)	(40%) (60%)	*p* = 0.051
BMI <18.5 18.5–25 25–29.9 30+	(5%) (35%) (31%) (29%)	(9%) (39%) (28%) (24%)	*p* = 0.004

**Table 2 vaccines-11-00475-t002:** The infectivity rate at each specific period with each specific vitamin D level.

	(25(OH)D) > 30	(25(OH)D) 20–30	(25(OH)D) < 20	Overall
3–4 months	1.20%	3.96%	3.26%	3.1%
5–6 months	1.47%	1.54%	2.86%	2.2%
7–8 months	1.78%	2.33%	3.58%	2.9%
9–10 months	4.70%	3.23%	2.54%	3.1%
11–12 months	7.15%	7.14%	5.38%	6.2%
3–12 months	2.64%	3.14%	3.36%	3.2%
3–8 months	1.53%	2.54%	3.34%	2.7%
9–12 months	5.39%	4.50%	3.39%	4.3%

**Table 3 vaccines-11-00475-t003:** Baseline characteristics of the recovered participants stratified according to anti-spike antibody titer <or> 50 AU/mL. CVA—cerebrovascular accident; CHF—congestive heart failure; PVD—peripheral vascular disease; IHD—ischemic heart disease; HTN—hypertension; DM—diabetes mellitus; COPD—chronic obstructive pulmonary disease; SES—socioeconomic state; BMI—body mass index.

	Anti-Spike Antibody < 50 (*n* = 1006)	Anti-Spike Antibody ≥ 50 (*n* = 2345)	Total (*n* = 3351)	*p*-Value
Age	44.9 ± 19.0	43.7 ± 17.6	44.05 ± 18.1	0.071
Gender Male Female	(35.1%) (64.9%)	(34.0%) (66.0%)	(34.4%) (65.6%)	0.55
Ethnicity Arab Orthodox Jewish Other	(34.6%) (14.9%) (50.5%)	(36.6%) (15.4%) (48.1%)	(36.0%) (15.2%) (48.8%)	0.43
25(OH)D level 0–20 20–30 >30	(46.0%) (37.2%) (16.8%)	(50.2%) (34.4%) (15.4%)	(48.9%) (35.2%) (15.8%)	0.093
PCR positive	(4.5%)	(2.5%)	(3.1%)	0.02
Anxiety	(38.6%)	(41.3%)	(40.4%)	0.26
Schizophrenia	(1.4%)	(2.6%)	(2.2%)	0.11
Depression	(22.2%)	(23.0%)	(22.7%)	0.73
Dementia	(1.2%)	(1.8%)	(1.6%)	0.45
Nephrotic syndrome	(0.8%)	(0.5%)	(0.6%)	0.53
Chronic Renal failure	(5.4%)	(4.5%)	(4.8%)	0.38
CVA	(3.0%)	(4.1%)	(3.8%)	0.21
CHF	(2.3%)	(2.5%)	(2.4%)	1.00
PVD	(3.4%)	(3.5%)	(3.5%)	1.00
IHD	(7.8%)	(6.5%)	(6.9%)	0.30
Hyperlipidemia	(53.0%)	(51.3%)	(51.9%)	0.50
HTN	(1.1%)	(1.4%)	(1.3%)	0.68
DM	(12.1%)	(10.2%)	(10.8%)	0.19
COPD	(7.5%)	(6.9%)	(7.1%)	0.64
Asthma	(20.2%)	(17.9%)	(18.7%)	0.22
SES 1–10 10–20	(59.5%) (40.5%)	(62.7%) (37.3%)	(61.7%) (38.3%)	0.10
Smoking status Active smoker Nonsmoker Former smoker	(9.9%) (88.5%) (1.5%)	(8.5%) (90.1%) (1.4%)	(8.9%) (89.7%) (1.4%)	0.44
BMI 16.5–18.5 18.5–24.9 25–29.9 30+	(3.7%) (31.1%) (35.1%) (30.1%)	(3.1%) (35.3%) (31.9%) (29.7%)	(3.3%) (34.0%) (32.8%) (29.8%)	0.11

## Data Availability

This study is based on real-world patient data, including demographics and comorbidity factors, that cannot be communicated due to patient privacy concerns.

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
