# Peer review of "SARS-CoV-2 Infection-Blocking Immunity Post Natural Infection: The Role of Vitamin D"

_vaccines, 2023, doi:10.3390/vaccines11020475_

Round 1

Reviewer 1 Report

This study examined the correlation between baseline 25(OH)D levels and both the rate of reinfection and anti-spike protein antibody titer following COVID-19 infection. The study concluded that higher pre-infection 25(OH)D levels can provide some protection against COVID-19 during the first 8 months after infection, although this protection does not seem to persist after this period.

However, the study has several problems that need to be addressed. First, the details of the Vitamin D assay used and when it was measured are not described. Second, when were the serology tests done, and if any of them were negative, was a retest done? Third, was all the subjects unvaccinated? If so, this should be clearly stated in the manuscript. Fourth, the figures presented in the paper are very confusing and difficult to interpret. Fifth, a description of the statistical analysis used is missing. Finally, there is no description of the ethical considerations taken into account when conducting the study.

These issues should be addressed and clarified to ensure that the results of the study are accurate and reliable. Additionally, the authors should provide a comprehensive explanation of the methodology used and the results obtained, as well as a clear description of the ethical considerations taken into account when conducting the study.

Author Response

Answers to Reviewer 1-

This study examined the correlation between baseline 25(OH)D levels and both the rate of reinfection and anti-spike protein antibody titer following COVID-19 infection. The study concluded that higher pre-infection 25(OH)D levels can provide some protection against COVID-19 during the first 8 months after infection, although this protection does not seem to persist after this period.

However, the study has several problems that need to be addressed.

First, the details of the Vitamin D assay used and when it was measured are not described.

Answer- Vitamin D measurement was mentioned in the head of page 8: "Notably, the median interval that elapsed from the time of 25(OH)D measurement to the time of the first documented infection was 4 months (IQR: 1-7)".

The Vitamin d assay used was specified in the head of page 7: " According to LHS guidelines, vitamin D tests were collected after overnight fasting, and transported on ice to the center laboratory for processing within 4 hours of collection using DiaSorin Chemiluminescence assay [23, 24, 25, 26]".  

According to LHS guidelines, vitamin D tests were collected after overnight fasting, transported on ice to the center laboratory for processing within 4 hours of collection using DiaSorin Chemiluminescence assay [23, 24, 25, 26]

  1. French D, Gorgi AW, Ihenetu KU, Weeks MA, Lynch KL & Wu AH (2011) Vitamin D status of county hospital patients assessed by the DiaSorin LIAISON® 25‐hydroxyvitamin D assay. Clin Chim Acta 412, 258–262.
  2. Rosecrans R & Dohnal JC (2014) Seasonal vitamin D changes and the impact on health risk assessment. Clin Biochem 47, 670–672.
  3. Moure Z, Rando‐Segura A, Gimferrer L, Roig G, Pumarola T & Rodriguez‐Garrido V (2018) Evaluation of the novel DiaSorin LIAISON. Enferm Infecc Microbiol Clin 36, 293–295.
  4. Thuzar M, Young K, Ahmed AH, Ward G, Wolley M, Guo Z, Gordon RD, McWhinney BC, Ungerer JP & Stowasser M (2020) Diagnosis of primary aldosteronism by seated saline suppression test‐variability between immunoassay and HPLC‐MS/MS. J Clin Endocrinol Metab 105, e477–e483. 

Second, when were the serology tests done, and if any of them were negative, was a retest done?

Answer- Figure 4 shows that the serology tests were done at different time intervals after recovery: 1,611 at 3-4 months post recovery, 851 at 5-6 months, 491 at 7-8 months, 225 at 9-10 months, and 174 tests done at 11-12 months post recovery.

The tests were done just once. No retesting was done if the first was negative, or even positive in attempt to explore dynamics…

Third, was all the subjects unvaccinated? If so, this should be clearly stated in the manuscript.

Answer- All subjects were unvaccinated. It is mentioned in page 6: "1) Recovery from documented COVID-19 infection between February 1, 2020, to January 30, 2022, in the absence of prior vaccination". It is also mentioned in the results section, page 8: " …. and without previous vaccination".

Fourth, the figures presented in the paper are very confusing and difficult to interpret.

Answer- relevant figures` legends were clarified.

Fifth, a description of the statistical analysis used is missing.

Answer- the statistical analysis section was added in pages 7&8.

Finally, there is no description of the ethical considerations taken into account when conducting the study.

Answer- the ethical consideration was attached in pages 15&16: " Institutional Review Board Statement: The study protocol was approved by Shamir Medical Center Institutional Review Board (013-21-LEU). This study was conducted in accordance with the Declaration of Helsinki".

"Informed Consent Statement: The need for informed consent was waived as part of the ethics approval of our study due to the retrospective design and low risk to the subjects".

Additionally, the authors should provide a comprehensive explanation of the methodology used and the results obtained, as well as a clear description of the ethical considerations taken into account when conducting the study.

Answer- a more comprehensive explanation of both the methodology used and the results obtained was provided in pages 5-10.

Reviewer 2 Report

The current study seeks to explain the importance of pre-infection levels of vitamin D on reinfection during convalescence and the limitation of immunoglobulin titers to serve as a stand-alone criterion to predict protective immunity against COVID-19. Vitamin D seems to play an important role in multiple diseases as it increases the overall diversity of the gut microbiota. I think that it would be interesting if the author could give a short discussion on this matter in connection with his findings. In any case, the is a gap in information on the role of vitamin D in COVID-19 infection and it should be of interest to get this knowledge on the subject.

The study embedded a retrospective observational survey well-designed. LHS where the study was conducted, possess a complete computerized database regarding the demographics, medical diagnoses, hospitalizations, vaccinations, and laboratory tests of patients. In this vein, 10,132 subjects fulfilling the inclusion criteria comprised the study cohort. Inclusion criteria required at least one available 25(OH)D level prior to enlistment Yet, they evaluated the rate of breakthrough infection and the anti-spike antibody level, subjects were divided into two groups according to anti-spike protein antibody titer. It was found a significantly higher proportion of these patients with anti-spike antibody <50  among positive RT-PCR subjects.

It is a well-written paper with all recent bibliographies in the field.

The study design is well presented, and the evaluation of the results is extensively discussed and correlated with the different aforementioned parameters.

In my opinion, it is a study that merits publication as it gives precious information about the role of vitamin D  efficacy, strengthens, and advances our knowledge in the field.

Due to the little knowledge that we have on the COVID disease and involving factors and parameters, this study merits publication.

I do not find grammatical errors throughout the study.

Author Response

Thank you for reviewing our manuscript

Round 2

Reviewer 1 Report

no further comments.

All previous comments were addressed except for the figures.

I still think they need to be better presented